# Resilience amid chaos: The role of Gaza medical points

**Ali Alshawwaf[1], Omar AlNajjar[2], Iyad Sultan[3,4], Eyad Qunaibi[5]***

1 Faculty of Pharmacy, Al-Azhar University, Gaza, Palestine, 2 Faculty of Medicine, Islamic University, Gaza City, Palestine, 3 Department of Pediatrics, Artificial Intelligence and Data Innovation Office, King Hussein Cancer Center, Amman, Jordan, 4 Faculty of Medicine, University of Jordan, Amman, Jordan, 5 MedOne Academy, Delaware, United States of America

* eyadq@medone.academy

## Abstract

The Gaza healthcare system faces critical challenges due to ongoing war, with medical points (MPs) serving as essential healthcare providers. MPs are temporary healthcare service units established during emergencies or conflicts to provide essential medical care. This study assessed the operational status and resource availability of MPs in Gaza during wartime conditions. A cross-sectional online self-assessment survey conducted from October to December 2024 evaluated 28 MPs across Gaza. Data on facility characteristics, services, equipment, drug availability, and staffing were analyzed using descriptive statistics. The findings revealed that MPs experience severe shortages in essential medications, with insulin, antiepileptics, and cancer treatments unavailable in over 90% of cases. Antibiotics and psychiatric medications are critically deficient. Equipment shortages, including limited oxygen supplies and diagnostic tools, further strain trauma and chronic disease care. Only 39% of MPs provide maternity and vaccination services, while mental health services are nearly absent. Staffing is inadequate, with many healthcare personnel lacking training. Most MPs operate in temporary structures with inconsistent electricity, internet, and sanitation. Despite these limitations, MPs (median staff of 7 personnel) manage an average of 117 patients per unit daily. These findings highlight that MPs are crucial in Gaza's healthcare delivery but face severe systemic challenges. Urgent action is required to ensure the uninterrupted supply of essential medications and medical equipment to sustain life-saving services. Additionally, telemedicine could help address access barriers and support healthcare providers in managing escalating demands.

## Introduction

The healthcare system in Gaza has long faced persistent challenges, compounded by prolonged conflicts and systemic underfunding. "A 2009 study published in the

**Data availability statement:** The data underlying this article are available upon reasonable request from the Institutional Review Board at MedOne Academy, USA (email: irb-committee@medone.academy). This body is independent from the study authors and is responsible for research oversight and data governance. Data access will be granted for academic, non-commercial purposes in accordance with participant confidentiality and ethical approval terms.

**Funding:** The author(s) received no specific funding for this work.

**Competing interests:** The authors declare that no competing interests exist.

*Canadian Medical Association Journal (CMAJ)* [1] highlighted the pre-existing strain on Gaza's healthcare infrastructure even before the 2008–2009 war. This strain was further exacerbated by repeated military attacks (2008–09, 2012, 2014, and 2021), sustained siege, and funding shortages, which have collectively restricted access to medical supplies and undermined healthcare services in Gaza prior to the current escalation.

Building on this, the ongoing military assault on Gaza that began in 2023 has had devastating effects on the district's medical infrastructure, resources, and personnel, pushing the healthcare system to the brink of collapse *as documented by the Office of the High Commissioner for Human Rights (OHCHR)* [2]. This assault has involved direct violence against healthcare workers, disruption of service delivery, and the criminalization, arrest, and detention of both patients and staff. As of May 22, 2025, 720 attacks on healthcare had been reported in Gaza, resulting in 917 deaths, around 1,000 injuries, and the detention of 300 health workers and 70 patients [3].

In the same month, 18 hospitals were partially functional across Gaza, with 18 of 36 hospitals completely out of service. Additionally, 8 field hospitals were operational, including five fully and three partially functional hospitals in various locations. At the same time, 39% of primary health care centers were functional (most of them partially), and only 33% of UNRWA health centers remained operational [4].

With ongoing evacuation orders issued by Israeli forces and continuous waves of displacement, approximately 1.9 million people have been displaced by May 2025, many displaced multiple times [5]—intensifying their vulnerability, disrupting access to consistent healthcare, and complicating the management of pre-existing conditions. In response to these dire circumstances, mobile medical teams and makeshift medical points have emerged as vital solutions to bridge the healthcare gap. Medical points, often deployed by local and international organizations, have been essential in providing healthcare in areas where conventional health facilities are either non-operational or overwhelmed. These points offer a range of services, including primary health care, trauma care and in some cases, maternal and child health services. Given the high number of internally displaced people in Gaza, the medical points play an even more crucial role, providing healthcare in temporary shelters and at sites where people are living in dire conditions [6].

Before the ongoing 2023 war, Gaza Strip maintained a high level of vaccination coverage across the population. However, the war significantly disrupted healthcare services causing routine immunization coverage for the second dose of inactivated polio vaccine to decline from 99% in 2022 to less than 90% in the first quarter of 2024. This decline heightened the risk of vaccine-preventable diseases, including polio, culminating in its reemergence despite a 25-year absence of reported cases [7]. Similar patterns have been observed in other conflict zones, such as Yemen, where El Bcheraoui et al. [8] demonstrated that sustained violence and systemic disruptions led to marked declines in vaccination coverage and the resurgence of vaccine-preventable diseases such as measles and cholera.

In response, an extensive vaccination campaign was launched. Medical points played a crucial role in reaching hard-to-access areas, ensuring that families unable to visit fixed sites due to security conditions were still vaccinated [9].

Publications and reports by organizations such as Norwegian Refugee Council and the United Nations Relief and Works Agency (UNRWA) have highlighted the crucial role of medical points in addressing urgent healthcare needs, ranging from trauma care to disease prevention and mental health services [6,10]. However, the volatile security situation and ongoing attacks on healthcare facilities undermine the stability of these services, making the provision of continuous care extremely challenging.

For all the reasons outlined above, a thorough understanding of the role, challenges, and needs of medical points is critical to informing sustainable healthcare strategies in conflict settings like Gaza. While previous literature has documented healthcare adaptations in other conflict zones, a detailed account of the medical point (MP) model is absent. In Yemen [11], mobile medical teams were deployed to manage cholera and acute watery diarrhea, and rehydration points were established within public healthcare facilities to address dehydration cases. In Ukraine [12], MPs were set up at border crossings with Poland to provide first aid and emergency care to fleeing civilians. In Syria [13], repeated airstrikes prompted the establishment of multiple smaller medical points instead of rebuilding large hospitals, aiming to enhance resilience and avoid total service disruption with single airstrikes. However, none of these studies addressed the operational status, functionality, or resource availability of MPs. This gap in the literature underscores the novelty and necessity of our study.

## Methodology

### Ethics statement

Ethical approval was obtained from the Ministry of Health in the State of Palestine. Electronic informed consent from participants was obtained through Google Form as a first page. Data confidentiality was maintained throughout the study.

### Sampling frame and recruitment

Using the Ministry of Health (MoH) master roster (version 18 Sep 2024), all 44 functioning MPs were enumerated. Study links were distributed by SMS to the telephone number registered for each MP and reminders were sent up to 3 times after initial contact.

Data collection was conducted over a three-month period, from October to December 2024, through an online self-assessment survey that was published using google forms. Of note, the MoH kept connectivity with the groups through a large WhatsApp group that was used to disseminate instructions and to connect MPs to facilitate patient transfer when necessary.

### Survey development

The Google Forms platform was selected due to its accessibility and user-friendly interface, which facilitated efficient data gathering [14,15]. Investigators reviewed the literature and thoroughly discussed all questions during study preparation. As two coinvestigators worked closely with MPs, areas that needed to be covered were recognized and addressed. We modelled many questions on the established WHO emergency health resource evaluation framework. The WHO Resources and Services Availability Monitoring System (HeRAMS) gathers information on essential health resources and services in crises [16]. The content covered infrastructure, safety, equipment, drug availability, and staffing, aligning with domains identified in previous field assessments.

### Variables and measures

All items were self-completed by a senior MP clinician; no external validation visit was possible.

1. **Facility Characteristics**: Data on location, catchment area, and structural types (permanent buildings, tents, mobile units) were collected. Security assessments included reports of previous attacks and perceived safety levels.

2. **Services and Equipment:** MP staff were asked about their capacity to deliver emergency care, primary care, maternity care and chronic disease management. The availability of diagnostic tools and emergency supplies was documented.

3. **Drug Availability**: Participants rated the availability of medications in 10 therapeutic categories, including antibiotics, antipyretics, antihypertensives, and insulin, using a five-point scale: 1 = not available (no drug in category available), 2 = available (1 or 2 medications), 3 = half of needed medications available, 4 = most medications available (1 or 2 missing), and 5 = all needed medications available..

4. **Staffing**: The composition, qualifications, and working conditions of MP staff were recorded. Participants were asked about their working hours, psychosocial support and training opportunities.

5. Infrastructure and utilities were evaluated with a five-point Likert scale reproduced verbatim from the questionnaire. For electricity and internet, respondents selected 1 = not available (0% of the time), 2 = rarely available (< 10%), 3 = sometimes available (10–50%), 4 = usually available (> 50%), or 5 = available all the time (≈ 100%). The same anchors were applied to psychosocial-support availability, where 1 indicated no support and 5 indicated comprehensive, continuous support. In addition, yes/no items captured the presence of functioning sewage drainage, hand-washing facilities (soap or alcohol-based rub), solid-waste disposal, and a refrigerator suitable for temperature-sensitive medications or vaccines.

6. All participants responded anonymously. The survey did not collect personal identifiers (names or contact information), ensuring confidentiality and protecting the safety of respondents in this conflict setting. Each medical point provided a single response to the survey. Responses are understood to reflect the situation during the data collection period (October–December 2024), as no specific reference period was indicated in the questionnaire.

### Data analysis

Quantitative data were analyzed using descriptive statistics, including frequencies, medians, ranges, and interquartile ranges. Data was analyzed using R program (version 4.4.1). Likert and ggplot2 were used to make figures. For workforce analysis, the non-numeric response "More than 5" was recoded as 6 to allow inclusion in staff totals and medians.

### Results

The response rate among contacted medical points in Gaza was 28/44 (63.6%). In many situations, failure to respond was due to the lack of internet connectivity, as reported by medical teams during follow-up phone calls. Some respondents had to move to another area to fill the survey as they had no connection at their locations.

A comprehensive analysis was conducted on the data from responding medical points. This dataset details operational characteristics, location, safety assessments, equipment availability, and drug supply levels. Approximate locations of participating MPs are shown in Fig 1.

Out of all surveyed medical points, 10 (36%) were explicitly identified as part of the Ministry of Health's operational network. NGOs (n = 15, 53.6) and local voluntary teams (n = 3, 10.7%) run other points. Twenty-six (92.9%) of the medical points were situated within areas designated as humanitarian safe zones. Four out of 28 medical points (14.3%) reported at least one staff member being injured or arrested during the period. One worker was killed in an airstrike. Place of injury or death was not specified in the survey. Two (7.1%) MPs were attacked with significant infrastructural damage.

Staffing levels varied by unit (Fig 2), Across the 28 responding medical points (MPs) we documented 68 doctors, 60 nurses, 28 paramedics, 30 pharmacists, and 62 other staff, for an overall workforce of 248 personnel (median 7 per MP,

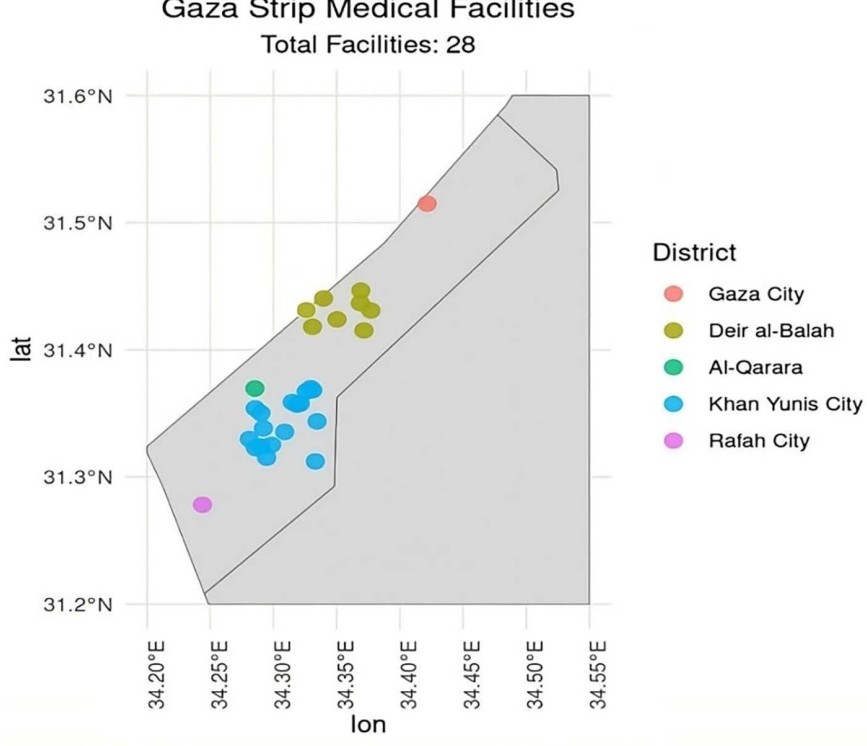

**Fig 1. Geographic distribution of medical points across Gaza Strip districts.** Points represent locations of medical facilities (n = 28) surveyed from Oct 2024 to Dec 2024. Each point represents one facility, jittered around its district center for clarity. Districts are ordered from north to south: Gaza City (n = 1), Deir al-Balah (n = 8), Al-Qarara (n = 1), Khan Yunis City (n = 17), and Rafah City (n = 1).

IQR 5–10, range 2–20). Medians by cadre were 2 doctors, 2 nurses, 0 paramedics, 1 pharmacist, and 2 support staff per MP, and three-quarters of units operated with no more than two physicians while 61% had no paramedic on site. As for training, 67.9% of workers (n = 19) reported receiving no training since joining their units. Psychological support of the staff was inadequate, with two-thirds reporting no support (n = 12, 42.9%) or minimal support (n = 8, 28.6%). Salaries were paid on time in 12 units (42.9%), with four units (14.3%) experiencing minor delays, while the remaining units reported inconsistent payments, including two units where no salaries were received.

Most units provided primary care (n = 27, 96.4%), pediatric care (n = 25, 89.3%), and care for chronic diseases (n = 25, 89.3%). Less than half of the units provided maternity care (n = 11, 39%) or emergency trauma care (n = 13, 46%). Dermatological, orthopedic, and ENT services were available in only one MP. Mental health services were available in only one other MP, as well. Some of the MPs hosted special programs for health education (n = 16, 57.1%), sexual and reproductive health services (n = 6, 21.4%), vaccination campaigns (n = 11, 39.3%), nutrition programs (n = 14, 50.0%), and prenatal care (n = 5, 17.9%).

Regarding facility and equipment availability, the analysis found that MPs had a median number of beds of 2. Basic diagnostic tools such as stethoscopes and blood pressure devices were present at all points, while advanced diagnostic equipment like portable ultrasound machines were available at only 5 points (17.9%). Emergency response capabilities were constrained, with oxygen supplies present in only 4 points (14.3%). Essential tools, including pulse oximeters (n = 11) and working glucometers with strips (n = 11), were available in less than half of the points.

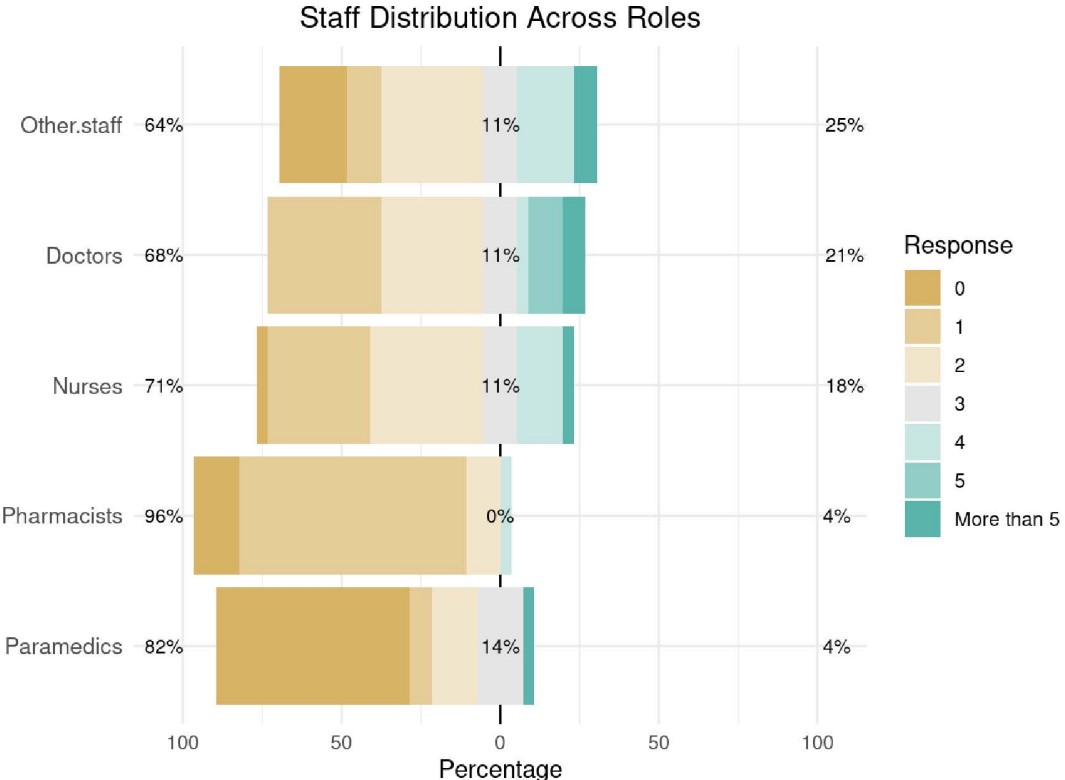

**Fig 2. Staff Distribution Across Roles in Medical Points.**

Although most MPs reported receiving cases of trauma regularly, suture kits were not available in one-third (n = 9, 32.1%). Similarly, essential supplies were lacking from a significant number of surveyed MPs, including sterile gauzes (n = 9, 32.1%), local anesthetics (n = 16, 57.1%), disinfectants like alcohol, povidone iodine, or chlorhexidine (n = 9, 32.1%). The majority had access to IV administration kits (n = 23, 82.1%), IV fluids (n = 24, 85.7%) and cannulas (n = 23, 82.1%). Nebulizers were missing in 8 MPs (28.6%). Only 2 (7.1%) had a working refrigerator.

As for medication availability, the mean availability score for antibiotics was 1.8 (SD = 1.4), with 12 MPs (42.9%) reporting no availability. Analgesics had a mean availability of 2.2 (SD = 1.4), with 13 points (46.4%) scoring 3 or higher. Cardiovascular drugs, including antihypertensives, were relatively more available, with 18 MPs reporting access to half or more of the needed medications. Drugs that showed the worst shortages included respiratory medications, psychiatric medications, antiepileptic drugs and cancer drugs (Fig 3). Antidiabetic medications had limited availability with only 2 (7.1%) of the MPs reporting any insulin supply.

MPs were hosted in tents and other temporary structures in 19 (67.9%) of the surveyed facilities. Electricity availability varied significantly, with 15 MPs (54%) reporting it as unavailable, rarely available (<10% of the time), or sometimes available (10–50% of the time). Internet connectivity followed a similar trend, with 6 MPs (21.4%) reporting full availability but 10 MPs (35.7%) reporting no or rare access. Sewage drainage was available only for 5 facilities (17.9%). Essential cleaning supplies were lacking in a significant number of MPs, including hand sanitizers (lacking in 8, 28.6%), soap (lacking in 9, 32.1%) and detergents (lacking in 18, 64.3%).

The working hours varied across units, with an average daily operation of 6.1 hours and most units operating six days per week (n = 24, 85.7%). On average, 117 patients were seen daily (SD = 44.6; median = 100; range 50–200).

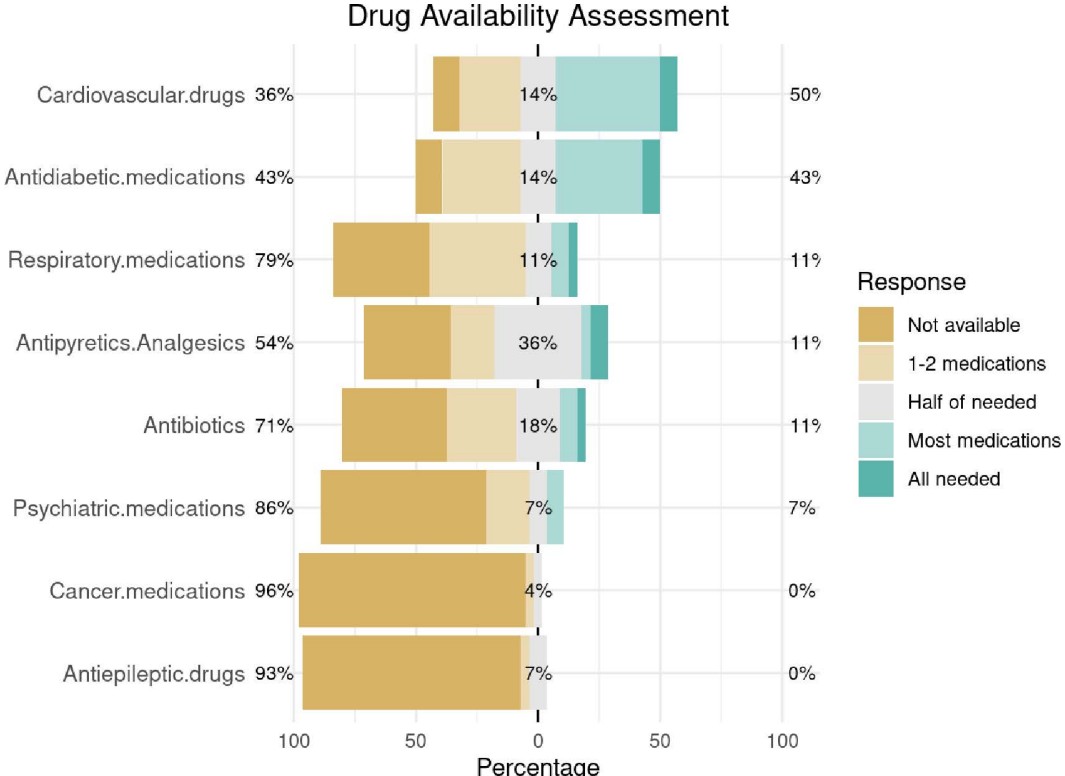

**Fig 3. Likert scores depicting drug availability across eight medication categories. Bars represent the distribution of responses (from "Not available" to "All needed medications available") among medical points in Gaza (n = 28). Survey was conducted from Oct 2024 to Dec 2024. Percentages to the left represent the sum of "Not available" and "1-2 medications available" and the percentages to the right represent the sum of "Most medications available" and "All needed medications available".**

Comprehensive patient records were maintained by only nine units, with three using electronic systems and six relying on detailed paper documentation. Brief records were kept by 12 units (42.9%), while seven either maintained no records (n = 1, 3.6%) or only recorded patient names (n = 6, 21.4%).

## Discussion

The results of our study on medical points (MPs) in Gaza reveal a healthcare system under extreme stress, struggling to provide even basic services amidst the ongoing assault. This crisis exacerbates an already precarious situation that existed before the October 7, 2023, war. Prior to this escalation, access to and affordability of healthcare services were significant challenges, with 70% of participants in a 2023 study focused on the elderly population reporting difficulties in these areas [17]. Our findings underscore critical shortages in essential medications, equipment, and infrastructure, painting a grim picture of healthcare delivery in a war-torn region.

The severe shortage of antibiotics observed in our study is particularly alarming, with 71% of MPs reporting no or very limited availability. This shortage is occurring in a context where Gaza is already experiencing a critical health crisis among children under 5 years, characterized by a significant rise in infectious diseases, including acute respiratory infections, diarrhea, scabies, and hepatitis A [18,19]. This crisis is driven by overcrowded living conditions, inadequate access to clean water and sanitation, and a critically strained healthcare infrastructure facing severe shortages of essential

resources and electricity in the Gaza strip as a whole [18]. The lack of sewage drainage, detergents, soap, and hand sanitizers was also evident within the MPs themselves in our study, further hindering efforts to combat infections.

The severe shortage of psychiatric medications reported here (>90% limited or unavailable) is alarming, particularly given the previously reported prevalence of 22% for conditions such as depression, anxiety, post-traumatic stress disorder, bipolar disorder, or schizophrenia among individuals affected by war or other conflicts [20]. International guidelines recommend various activities for providing mental health and psychosocial support (MHPSS) during emergencies, ranging from community self-help and communications to psychological first aid and clinical mental health care. A recent review offered insights for improving mental health and psychosocial support (MHPSS) for Gaza's populations [21]. Our findings on critical psychiatric medication shortages are particularly alarming because they signify an acute disruption of essential treatment for individuals already affected or at high risk of developing mental health conditions, especially within a context of pervasive trauma and loss. This medication crisis is further compounded by the extremely limited availability of mental health services in medical points as shown in our study, with only 7% offering such care, leaving a profound unmet need for mental health care across key recommended intervention layers.

A similar concern arises from the severe shortage of antiepileptic drugs, particularly given the reported worsening of epilepsy and the heightened psychiatric problems faced by epileptic patients during war [22]. The near-total absence of cancer medications reported here is further exacerbated by the documented shortage of palliative medicines in the Gaza district [23].

To the best of our search, we did not find published data quantifying the extent of medication shortages in Gaza's hospitals, field hospitals, or primary healthcare centers. However, the situation may be comparably severe, as UNRWA warned in December 2024 that at least 60 medications would be depleted at its health facilities by year's end [24]. Therefore, the overall medication shortage in Gaza may exceed that reported in other conflict zones, such as Yemen, where approximately half of medications were available [25]. This shortage undermines the management of both acute and chronic conditions, potentially resulting in severe health complications for the affected population.

The lack of maternity and prenatal care in MPs, as evidenced in our study, jeopardizes maternal health and endangers the long-term well-being of future generations. This is further compounded by the low availability of vaccination services, which has contributed to the reemergence of polio and increased the risk of other vaccine-preventable diseases [7].

Despite the presence of various healthcare professionals in MPs, our study reveals critical challenges that hinder their functionality. With 67.9% of workers reporting no training since joining their units, there is a risk of skill stagnation in a rapidly evolving crisis situation. This lack of continuous education could impact the quality of care provided, especially in managing complex cases arising from the assault.

In addition, healthcare personnel operating in MPs face numerous challenges, including physical harm, lack of psychological support, and irregular salary payments. Addressing these systemic issues is crucial to ensuring the sustainability and efficacy of these frontline workers. These challenges are further compounded by extremely high workloads, with each unit, staffed by a median of seven personnel, managing an average of 117 patients daily.

The severe shortage of critical medical equipment, including oxygen supplies, portable ultrasound machines, suture kits and sterilization materials, severely limits the capacity of medical points to deliver adequate care, particularly for trauma and critical patients. With the rising number of such cases during the assault on Gaza, the absence of these supplies is likely contributing to preventable deaths. While respondents detailed the availability of critical supplies, the survey did not examine how staff coped during stockouts—an important operational gap that warrants future qualitative investigation.

The blockade on Gaza—already in place for over 16 years—had severely under-resourced the health system even before the current war. Since October 7, 2023, the crisis has been compounded, turning the blockade into a critical barrier to medical response by restricting the sustained entry of essential medicines and equipment [26]. Médecins Sans Frontières reported a continued inability to bring medical supplies into Gaza since April 2024, warning of severe shortages that left patients with burns, open fractures, and other traumatic injuries without basic painkillers, and forced surgical teams

to operate without anesthetics [27,28]. The blockade has thus become a lethal determinant, accelerating the collapse of healthcare infrastructure and rendering even basic care unattainable.

Two-thirds of the MPs were hosted in tents and other temporary structures. While mobile clinics are often recommended and utilized to reach those without access to traditional facilities, the literature lacks sufficient peer-reviewed data on decision support tools to optimize their deployment [29]. Some practical lessons can be drawn from war-affected regions like Syria, where new health facilities were deliberately built away from town centers to minimize civilian casualties in case of airstrikes, and services were divided across separate locations to avoid total service disruption from a single attack [13]. While such decentralization improved access and resilience under bombardment, it also complicated coordinated care, highlighting both the utility and limitations of MPs in conflict settings.

Given the long-standing political paralysis surrounding Gaza's health crisis—despite repeated international appeals—our recommendations focus on practical measures that remain viable amid sustained inaction. Implementing a locally managed, transparent supply-tracking system could help prioritize critical shortages and enhance distribution efficiency. In parallel, task-shifting programs to train non-specialist staff in basic emergency and chronic care may partially alleviate the burden created by the ongoing loss and exhaustion of qualified personnel. These steps cannot address the structural roots of the crisis; without reliable humanitarian access to restore essential supplies, all solutions remain partial and palliative—insufficient to counter the obstruction of life-saving aid. Nonetheless, they represent actionable pathways to sustain medical point operations under extreme constraints.

Healthcare professionals (HCPs) outside Gaza can play a vital role in mitigating the catastrophic healthcare situation through telemedicine. By offering virtual consultations, diagnostic support, and mental health services, telemedicine can partially overcome physical barriers to healthcare in war zones [30]. It holds promise not only to address the mental health needs of Gaza's general population but also to support its overburdened healthcare staff, especially given that two-thirds reported receiving no or minimal psychosocial support.

Several organizations and volunteer teams have established online platforms, such as WhatsApp groups (WhatsApp, Meta; Menlo Park, CA, USA), For example, Gxza Health reports providing thousands of teleconsultations, psychosocial support, and coordinating medication delivery through WhatsApp-based outreach [31,32].One study on the use of telemedicine at Nasser Hospital in Gaza demonstrates how telemedicine support has emerged as a vital lifeline for healthcare workers on the ground, allowing for the provision of multispecialty advice for complex patient cases [32]. These findings underscore the potential for similar telemedicine approaches to be adapted and expanded within Gaza, including medical points.

However, while these efforts represent palliative attempts to bridge critical care gaps where formal systems are severely strained, they face several challenges. First, their effectiveness is profoundly hampered by Gaza's intermittent internet and electricity. Our findings indicate only 21.4% (n = 6) of surveyed medical points (MPs) reported full internet availability, and 35.7% (n = 10) had no or rare access, with consistent electricity also being a major issue. Additionally, public engagement has been slow due to the novelty of these initiatives and weak advertising. Moreover, the use of commercial messaging platforms for clinical purposes raises critical data protection, privacy, and security concerns. Although WhatsApp offers end-to-end encryption, it is not designed or certified for managing sensitive health data in conflict zones. Health and location information of both patients and health workers may be exposed to hostile interception. The 2022 cyberattack on the International Committee of the Red Cross's (ICRC) protected database [33] highlights the risks of relying on unsecured digital systems, and the urgent need for secure, low-bandwidth, contextually appropriate telemedicine platforms designed for conflict settings. In parallel, clinical support from external professionals must be not only technically competent but also culturally and linguistically aligned with Gaza's population. Our survey did not assess how services were communicated or advertised to the local community, which may affect utilization patterns and awareness of available care.

Despite these challenges, increasing positive feedback from beneficiaries has gradually improved participation, and these initiatives now represent a crucial step toward alleviating the healthcare crisis in Gaza.

Additionally, HCPs outside Gaza can support by developing decision support tools to optimize MP deployments, which could be implemented by healthcare professionals within Gaza to enhance their effectiveness.

## Limitations

While our study provides valuable insights into the functioning of MPs in Gaza, its findings are limited by the cross-sectional design and the volatile and rapidly changing situation on the ground. The response rate was 63.6% (28 out of 44 identified MPs), possibly introducing response bias. However, this rate was achieved under active displacement and infrastructure destruction conditions. During the study period, Gaza experienced severe communications disruptions – frequent internet blackouts and infrastructure damage often left communities cut off from communication [34]. Regarding data analysis, some staffing responses were non-numeric ("More than 5") and were conservatively recoded as 6, which may slightly underestimate total workforce counts. Furthermore, as the study focused exclusively on medical points, the findings may not be generalizable to other healthcare structures such as hospitals, field hospitals, or primary healthcare centers. Finally, although the study relied on self-reported data, responses were provided by healthcare workers directly involved in MP operations, offering firsthand and timely insight.

Importantly, the successful completion of this study during ongoing military assault and systemic infrastructure collapse underscores the feasibility of conducting field research in extreme conditions. Despite being a cross-sectional study, it offers a rare, practice-based contribution to the conflict-health literature by demonstrating that meaningful data collection is possible even under severe hardship.

## Conclusion

This study underscores the critical role of medical points in providing essential healthcare services in regions of severe humanitarian crisis like Gaza. Despite their limitations, these units serve as a lifeline for displaced populations, ensuring access to primary care and trauma management. However, the situation is highly alarming, with systemic challenges, including severe shortages in medications, resources, and equipment, as well as inadequate support for healthcare workers. These deficiencies jeopardize the health of vulnerable populations, exacerbate preventable mortality, and hinder efforts to address both acute and chronic health needs. Urgent action is required to address these systemic gaps and ensure sustainable healthcare delivery for Gaza's affected population. While the political actors are responsible for stopping this crisis, the medical and humanitarian community can contribute through immediate, actionable measures such as expanding telemedicine services, supporting remote training for Gaza-based providers, and offering psychosocial support to frontline healthcare staff.

## Supporting information

**S1 Data. Data collected from Gaza Medical Points.**
(XLSX)

## Author contributions

**Conceptualization:** Ali Alshawwaf, Omar AlNajjar, Iyad Sultan, Eyad Qunaibi.

**Data curation:** Ali Alshawwaf, Omar AlNajjar.

**Formal analysis:** Iyad Sultan.

**Methodology:** Ali Alshawwaf, Omar AlNajjar, Iyad Sultan, Eyad Qunaibi.

**Validation:** Iyad Sultan, Eyad Qunaibi.

**Visualization:** Iyad Sultan.

**Writing – original draft:** Omar AlNajjar, Iyad Sultan, Eyad Qunaibi.

**Writing – review & editing:** Eyad Qunaibi.

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
