## [Decision Letter · Decision Letter 0]

10 Apr 2025

PGPH-D-25-00404

Resilience Amid Chaos: The Role of Gaza Medical Points

Dear Dr. Qunaibi,

Thank you for submitting your manuscript to PLOS Global Public Health. After careful consideration, we feel that it has merit but does not fully meet PLOS Global Public Health’s publication criteria as it currently stands. Therefore, we invite you to submit a revised version of the manuscript that addresses the points raised during the review process.

We look forward to receiving your revised manuscript.

Kind regards,

Hani Mowafi, M.D., M.P.H.

Academic Editor

Journal Requirements:

Additional Editor Comments (if provided):

Reviewers' comments:

Reviewer's Responses to Questions

**Comments to the Author**

1. Does this manuscript meet PLOS Global Public Health’s publication criteria?

Reviewer #1: No

Reviewer #2: Partly

2. Has the statistical analysis been performed appropriately and rigorously?

Reviewer #1: No

Reviewer #2: Yes

3. Have the authors made all data underlying the findings in their manuscript fully available (please refer to the Data Availability Statement at the start of the manuscript PDF file)?

Reviewer #1: No

Reviewer #2: Yes

4. Is the manuscript presented in an intelligible fashion and written in standard English?

Reviewer #1: Yes

Reviewer #2: Yes

Reviewer #1: Thank you for this important and timely piece of work amidst a very trying time in Gaza. This study provides valuable insights into the role of medical points in Gaza amid the ongoing Israeli assault. While the research is well-structured and addresses a critical issue, several areas require refinement. Please see comments below for breakdown by section:

Introduction

- Please ensure to have relevant citations early (ex: line 64; this study is not cited until the end of the paragraph); furthermore

- Please ensure that acronyms are spelled out first prior to using them (ex: line 64, the use of CMAJ).

- Please have a citation for your statement that the healthcare system is on the brink of collapse

o I suggest using the UN report published in October 2024 regarding attacks on healthcare

- Very clear definition of medical points; might be good to have a citation pointing to their use in other settings previously

- With regards to disruption of vaccination services, may also be good to point to literature where war was seen as driver of drop off of vaccination (there is a study done in Yemen that highlights this). This may strengthen your assertion that the war is a major driver in the vaccination rates

Methodology

- Citation needed for Google Forms

- Was participant data anonymized?

- Please clarify if the data for each MP was extracted via one representative or if multiple workers within the MPs were able to fill out the surveys

- Please clarify how you were able to qualify “failure to respond was due to internet connectivity” versus other reasons for non-response

o This may also need a citation regarding availability of internet connectivity in the Gaza Strip during the study period

- Please clarify what literature was reviewed to develop the survey

- Please clarify how the scale of drug availability was determined; i.e. what does “fully available” mean, and how was this definition communicated to those who filled out the survey?

Results

- How many total MPs were reached out to for the study? You will need this for the denominator to see response rate.

- Please report the results consistently as follows: n (%)

- Please ensure the data is reported objectively, i.e. with numbers as opposed to with qualifiers such as “significant”; this may be done in the discussion

- Please clarify if in line 181 n=4 refers to the number of workers harmed or arrested, or if this refers to the number of MPs that had at least one worker harmed or arrested

- In line 183, please change “targeted” to “attacked”

- Definitions within the survey (ex: electricity availability in line 212-213), should be in the methods

Discussion

- The study in reference 7 appears to focus on the elderly, yet the statement is made more generally; please ensure other references that can speak more generally to this statement or revise the statement to reflect the population studied in the relevant citation

- There are statements within the discussion which overstate the results or make claims that are not within the scope of the study, such as in line 254 where you speak about shortage of medications exacerbating the health crisis.

o This occurs in line 275 where the findings of your study are generalized to the entire Gaza strip and is then compared to Yemen. It would be more appropriate to state that the medication shortage is observed within MPs, since hospitals and other static structures within Gaza are not included within the study.

- The discussion restates the findings of the study; this space should be used to discuss recommendations and make references to the literature about the state of healthcare in other conflict settings if literature on medical points is limited

- It may be useful to speak to the blockade that may affect the influx of critical medications and equipment in the Gaza Strip

Limitations

- This section can be expanded to include response rate, that the study only looked at MPs and its results may not be reflective of all medical structures in Gaza, among others

Conclusion

- Please specify “urgent action”

Reviewer #2: Thank you for the opportunity to review this important study on medical points in Gaza. The study not only describes the severe challenges faced by the health system, but it also demonstrates that it is feasible to do this kind of research in the midst of high intensity conflict, and then share the findings with the wider world to advocate. The study is an important contribution to the literature on conflict and health.

I have made a number of suggestions below. The most important suggestions relate to strengthening the methods, results and discussion. I hope this is useful.

Introduction:

Lines 65-68: Does the exacerbation of strain refer to the effect of the subsequent 5 wars and the siege? If so, please make this clear here.

Line 69: Suggest rewording to clarify this, something to the effect of “The ongoing military assault on Gaza that began in 2023…”

Line 72 – physical assault on who? Does this paragraph refer specifically to attacks on health workers or more generally? If it is about attacks on health care, please make this clear.

Line 75: suggest changing “divided by” to “including”

Lines 84-87: suggest to also state that many people have been displaced multiple times

*Please review the links in the references, some of these are not working.

Methods:

Please describe the sampling method. How many MPs exist in Gaza (is it 44)? Was every single medical point contacted via SMS?

First paragraph (lines 124-138): Please move the description of sampling to be placed before the description of data collection.

Lines 130-131: The sentence about failure to respond from lack of internet should be moved to the results. But please describe in the methods how the research team determined this – was a follow up made afterward the study? If so please add this to the methods. Or did health points reach out after the deadline for survey response? If so, this should also be described in the results, not the methods.

Variables and measures: This reads as if the facilities were assessed, but the design is limited to self-report by the MP staff. It is important to make this distinction.

• Point 1 is very clear.

• Point 2 suggests evaluation of capacity – this should be rephrased to something like “MPs were asked about their capacity to…”.

• Point 3 should be clarified. Were participants asked about medication supply for the 10 categories? If so, please reword to make this clear

• Point 4 is clear, but it would be improved if the second sentence were reworded to something such as “Participants were asked about their working hours, psychosocial support, and training opportunities.”

Please specify the time frame that the MPs were surveyed about. Was the survey about conditions since October 7, 2023? Or was it a different time period? What was the time frame for the questions on supply/stock?

Did the survey ask about how services were advertised to the community? For example, if trauma services were provided, how would patients know they could go to the MP rather than a hospital?

Could you please share the survey questions in supplementary materials, so readers can understand how the data were collected? I appreciate this would require translation to English but it would be very useful.

There is some repetition in the first and second sentences of this section.

Results:

Line 172 – suggest rewording to “The dataset included information from 28 medical points….”

Line 182: Were workers physically harmed while working in the medical point? Please specify this.

Lines 190-191: Were there strips for the glucometers? It is important to note that there is a supply requirement for these also, many public health professionals won’t be aware of that.

Line 192: Suggest removing “the fact” – this is based on participant report, and should be reported as such.

Did participants report on what they did when supplies ran out, for example suture kits, disinfectant or bandaging?

Please report on the numbers and training of staff that provide services in the MPs. Description of staffing as well as the kinds of services offered (lines 219-226) should be moved to early in this section, so the reader can understand the available resources for staffing as well as equipment.

Please rework the drug supply paragraph or share the Likert scale so the reported numbers can be clearly understood. Also, the findings show very limited cardiovascular medication availability – I appreciate there was better availability, but it seems like a bit generous to say it was “the best”. I suggest to word it slightly differently: “Cardiovascular drugs, including hypertensives, had slightly better availability, with 18 MPs reporting access to…”

Please report number (N) and percent for all statistics shared. At present it is variably reported, sometimes just a number, sometimes both number and percent.

Line 222: Were the mental health services provided by trained professionals? What kind of professional (doctor, psychologist, other paraprofessional such as a social worker or trained volunteer)? Also it reads they were available in “only one other MP” – how many MPs offered mental health services?

Were participants asked about who provided psychological support to them? Was it their families? Or friends? Or a formal service for health workers? This is very important.

Line 240 – please specify how long the MP was functioning with staff that were not receiving salary.

Discussion

The discussion focuses heavily on telemedicine but is much weaker on addressing the systemic issues. There are some studies on telemedicine now which help to describe the uses as well as challenges and risks – and it is only palliative because it doesn’t stop the underlying problems, such as limited staff, supply disruptions, lack of equipment, attacks on health care. The discussion would be much stronger if it also directly addressed some of these factors.

Line 256- is the jaundice seen in kids from hepatitis? Or could it also be from non-infectious causes?

Lines 255-256: please provide a citation for this sentence.

Lines 263-274: Recommendations for mental health services in humanitarian settings place emphasis on access to basic needs, strengthening community supports, and then smaller non-specialist services, and lastly, medications and specialist support. The focus here on medications implies the reverse and undermines the paper, as it gives the impression that mental health services are focused specifically on prescribing medication, when this is not standard of care even in peaceful and well-resourced settings. It would help to provide some context for why there is a focus on medications – was there a population that was already receiving medications prior to this war, and who are now no longer able to access their medicines? Or is there another reason for the emphasis on medication?

Lines 301-302: The study design does not allow for causal attribution. This statement should be amended, to “likely contributing to preventable deaths.” If there was an assessment of mortality, this should be reported in the results and discussed in the discussion.

Lines 307-312: This paragraph on telemedicine should be expanded further. Telemedicine is useful as a palliative measure when other solutions to improve resources are not possible. There are barriers identified in this review – only 21.4% (n=6) had uninterrupted internet access and 35.7% (n=10) had no or rare access. So telemedicine will not work well for a large proportion of the MPs unless internet access improves.

Also, telemedicine could also be used to provide mental health support for health workers, as well as the general population of Gaza.

Lines 313-323: were groups providing clinical services over WhatsApp? Or how was WhatsApp used? Please note that this violates all data protection laws and can place people in danger to both patients and health workers through health and location information becoming available and possible to intercept. The ICRC protection database was hacked in late 2022 – and that was a database that had significant protections put into place because it held sensitive information. (https://www.icrc.org/en/document/icrc-cyber-attack-analysis)

Lines 313-323: please provide citations for the different organisations and programmes referred to here and evidence for their success.

Lines 324-329 – please see my earlier note about WhatsApp. I appreciate this is handy and readily available but there are very real dangers to both patients and health workers with its use. These should be acknowledged, at the least. There is a need for readily available, secure and appropriate platforms for telemedicine as well as health workers outside of Gaza who understand the context and can provide telemedicine support.

Limitations – I think it is important to recognise that this study was carried out in spite of enormous challenges. The fact that it was even completed is a testament to the feasibility of doing research in areas experiencing high intensity conflict. Demonstrating this feasibility – even if it is simply a cross-sectional study – is still a huge contribution to the literature on conflict and health.

**Do you want your identity to be public for this peer review?** For information about this choice, including consent withdrawal, please see our Privacy Policy

Reviewer #1: No

Reviewer #2: No

---

## [Decision Letter · Decision Letter 1]

20 May 2025

PGPH-D-25-00404R1

Resilience Amid Chaos: The Role of Gaza Medical Points

Dear Dr. Qunaibi,

Thank you for submitting your manuscript to PLOS Global Public Health. After careful consideration, we feel that it has merit but does not fully meet PLOS Global Public Health’s publication criteria as it currently stands. Therefore, we invite you to submit a revised version of the manuscript that addresses the points raised during the review process.

We look forward to receiving your revised manuscript.

Kind regards,

Hani Mowafi, M.D., M.P.H.

Academic Editor

Journal Requirements:

Additional Editor Comments (if provided):

Please resubmit including the rebuttal to both reviewers comments. Only one set of comments were addressed in the attached documents.

Thank you

Reviewers' comments:

Reviewer's Responses to Questions

**Comments to the Author**

Reviewer #2: (No Response)

publication criteria?

Reviewer #2: (No Response)

3. Has the statistical analysis been performed appropriately and rigorously?

Reviewer #2: (No Response)

4. Have the authors made all data underlying the findings in their manuscript fully available (please refer to the Data Availability Statement at the start of the manuscript PDF file)?

Reviewer #2: (No Response)

5. Is the manuscript presented in an intelligible fashion and written in standard English?

Reviewer #2: (No Response)

Reviewer #2: (No Response)

**Do you want your identity to be public for this peer review?** For information about this choice, including consent withdrawal, please see our Privacy Policy

Reviewer #2: No

---

## [Decision Letter · Decision Letter 2]

9 Jul 2025

PGPH-D-25-00404R2

Resilience Amid Chaos: The Role of Gaza Medical Points

Dear Dr. Qunaibi,

Thank you for submitting your manuscript to PLOS Global Public Health. After careful consideration, we feel that it has merit but does not fully meet PLOS Global Public Health’s publication criteria as it currently stands. Therefore, we invite you to submit a revised version of the manuscript that addresses the points raised during the review process.

Please make the minor revision as suggested by reviewer 1 of adjusting Figure 1.

We look forward to receiving your revised manuscript.

Kind regards,

Hani Mowafi, M.D., M.P.H.

Academic Editor

Journal Requirements:

Additional Editor Comments:

Please adjust figure 1 as suggested by reviewer 1 and resubmit as soon as possible.

Reviewers' comments:

Reviewer's Responses to Questions

**Comments to the Author**

Reviewer #1: All comments have been addressed

Reviewer #2: All comments have been addressed

publication criteria?

Reviewer #1: Yes

Reviewer #2: Yes

3. Has the statistical analysis been performed appropriately and rigorously?

Reviewer #1: Yes

Reviewer #2: Yes

4. Have the authors made all data underlying the findings in their manuscript fully available (please refer to the Data Availability Statement at the start of the manuscript PDF file)?

Reviewer #1: No

Reviewer #2: Yes

5. Is the manuscript presented in an intelligible fashion and written in standard English?

Reviewer #1: Yes

Reviewer #2: Yes

Reviewer #1: Thank you for addressing the comments so thoroughly. My only comment is for figure 1 where one of the medical facilities appears to not be on land; please adjust this.

Reviewer #2: This paper is excellent. Thank you for your work, and for your perseverance to bring the paper to the wider public health community.

**Do you want your identity to be public for this peer review?** For information about this choice, including consent withdrawal, please see our Privacy Policy

Reviewer #1: No

Reviewer #2: No

---

## [Editor Report · Decision Letter 3]

1 Aug 2025

Resilience Amid Chaos: The Role of Gaza Medical Points

PGPH-D-25-00404R3

Dear Prof Qunaibi,

We are pleased to inform you that your manuscript 'Resilience Amid Chaos: The Role of Gaza Medical Points' has been provisionally accepted for publication in PLOS Global Public Health.

Best regards,

Hani Mowafi, M.D., M.P.H.

Academic Editor

Thank you for the revision